# Enhancing Student Knowledge of Diabetes through Virtual Choose Your Own Adventure Patient Case Format

**DOI:** 10.3390/pharmacy9020087

**Published:** 2021-04-20

**Authors:** Tyler Marie Kiles, Elizabeth A. Hall, Devin Scott, Alina Cernasev

**Affiliations:** 1Department of Clinical Pharmacy and Translational Science, College of Pharmacy, The University of Tennessee Health Science Center, Memphis, TN 38163, USA; liz.hall@uthsc.edu; 2Teaching and Learning Center, College of Pharmacy, The University of Tennessee Health Science Center, Memphis, TN 38163, USA; dscott50@uthsc.edu; 3Department of Clinical Pharmacy and Translational Science, College of Pharmacy, The University of Tennessee Health Science Center, Nashville, TN 37211, USA; acernase@uthsc.edu

**Keywords:** pharmacy education, teaching and learning, critical thinking, diabetes management

## Abstract

Educational strategies to teach pharmacy students about diabetes are necessary to prepare future pharmacists to manage complex patients. The Choose Your Own Adventure (CYOA) patient case format is an innovative activity that presents a patient case in an engaging way. The objectives of this study were (1) to describe the development of the innovative teaching activity and (2) to assess its effect on student knowledge and confidence in outpatient management of diabetes. The CYOA patient case activity was designed by transforming a traditional paper patient case involving outpatient diabetes management into an interactive format utilizing an online platform. The activity was conducted with 186 second-year pharmacy students in a skills-based course. This activity was administered virtually through a combination of small group work and large group discussion. After completion of the activity, students completed an online self-assessment questionnaire. Of 178 completed questionnaires, there was a statistically significant difference in students’ self-ratings after versus before the activity for all survey items (*p* < 0.001). The CYOA activity improved self-reported knowledge of outpatient diabetes management and increased self-reported confidence in clinical decision-making skills. This format shows promise as an educational tool that may be adapted for other disease states to enhance clinical decision-making skills.

## 1. Introduction

More than 34 million people in the United States have diabetes, and diabetes is the seventh leading cause of death [1]. A rise in the prevalence of diabetes is occurring worldwide in parallel with an increasing prevalence of obesity in children [1,2]. As the prevalence of diabetes increases, so does the need for appropriate medical management, as poorly controlled diabetes can lead to amputations, vision loss, and kidney damage. Interventions to help people manage diabetes can reduce the risk of these complications, and pharmacists can play a key role: from patient education and medication adherence, to monitoring and counseling on behavioral and lifestyle modifications. As new therapeutic options for diabetes become available and medication management becomes more complex, there is also an opportunity for pharmacists to help manage patients with diabetes as part of interprofessional healthcare teams [3,4,5].

Diabetes is a multifaceted disease state, and its therapeutic management can be challenging for pharmacy students to fully grasp. Students must master not only the complicated pathophysiology and ever-evolving pharmacotherapy options, but also synthesize this information to make clinical decisions for complex patients who often present with multiple comorbidities and social determinants of health [6]. Additionally, students must be able to consolidate and translate this information to express empathy, utilizing patient-centered communication techniques. Recognizing the need for mastery of this subject, many innovative educational approaches have been used to teach pharmacy students about diabetes. Several colleges of pharmacy have offered elective diabetes courses, certifications, and/or advanced simulations to meet this challenge [7,8,9,10,11,12]. These studies suggest that immersion in the diabetes experience improves the students’ ability to educate and empathize with diabetic patients. Underpinning the development of the aforementioned active-learning approaches is the understanding that student engagement in the learning experience is critical for learning about diabetes, and that lecture-based instruction may not provide students with a complete understanding of diabetes management. However, the available literature focuses largely on developing student empathy and patient education skills and does not assess the impact of educational techniques on knowledge and confidence with clinical decision making and the pharmacotherapeutic management of diabetes.

Clinical decision making and the effective pharmacotherapeutic management of diabetes are based on the critical evaluation and judgement of information; however, teaching this can be challenging to accomplish in the classroom. As a result, students often struggle to transition from understanding and interpreting information to applying these skills in experiential settings. While achieving immersion in the diabetic experience is challenging in the classroom setting, small-group teaching methods to improve problem solving skills—problem-based learning (PBL) and case-based learning (CBL)—have been well-studied [13]. PBL activities allow students to work through problems independently, devise solutions, and discuss with their peers. PBL has advantages for students, because open inquiry allows students to struggle and explore, directly developing problem-solving skills. However, due to curriculum density, students and faculty may prefer CBL, in which advanced preparation is required, and an “expert” guides students to a “correct” answer, developing clinical expertise. While PBL is student-centered, requiring students to engage in independent analysis, CBL is teacher-centered in that the expert leading the discussion knows the “answer” and provides guidance along the way. Based on combining the principles of CBL and PBL, the CYOA patient case format was developed. The CYOA activity encourages students to work through problems and make clinical decisions on their own, while simultaneously leveraging facilitation and discussion to guide students to arrive at a “correct” answer. The CYOA patient case format is an innovative activity that presents a patient case in a fun and engaging way.

The objectives of this study were (1) to describe the development of an innovative teaching activity and (2) to assess its effect on student knowledge and confidence in outpatient management of diabetes.

## 2. Materials and Methods

### 2.1. Description of the CYOA Diabetes Activity

A traditional paper patient case regarding outpatient diabetes management was transformed into an interactive CYOA patient case format utilizing a free online survey platform, QuestionPro (Dallas, TX, USA) The activity was designed to allow small groups of students to navigate through the patient case step-by-step, and was created to be visually appealing, incorporating color, images, and gifs to promote student engagement. Students were presented with parts of the patient case and had to make several choices (and provide rationale) along the way. Once students selected a clinical decision, students were routed (through survey logic) to another page describing the outcome of that choice. Each outcome provided a text explanation as to why or why not that decision was the most correct. While all choices were valid options, there was only one series of choices that eventually led to the most positive patient outcome. If students selected a suboptimal clinical decision, they were redirected back to select a different clinical decision. The CYOA case allowed students to make changes in medication management, select dietary recommendations, as well as practice carbohydrate counting, while providing feedback each step of the way. Students worked in small, virtual groups (using Zoom breakout rooms) and were facilitated by an upperclassman teaching assistant until coming to designated stopping points for large group discussion, led by the course instructor. The course instructor was a subject matter expert who holds a board certification in advanced diabetes management.

### 2.2. Data Collection and Analysis

This study was approved by the University of Tennessee Health Science Center (UTHSC) Institutional Review Board (IRB). The activity was administered over 50 min to 186 second-year pharmacy students in the Interprofessional Education and Clinical Simulation (IPECS) III Course at UTHSC College of Pharmacy. The students had already received 6 h of didactic lecture content related to diabetes before participating in the activity; however, students had not yet taken an examination on the material. The learning objectives of this activity were to implement appropriate clinical decisions regarding outpatient intensive insulin therapy and to understand the diabetic patient experience. This activity was delivered 4 days before the examination on the diabetes material. While attendance in the IPECS III course is mandatory, students did not receive a grade for participation in this activity.

After completion of the activity, students completed a brief online questionnaire. This 24-item survey was designed using the following domains: (1) student knowledge and confidence regarding patient case preparation, (2) critical thinking, and (3) activity learning objectives. The survey instrument was developed by the investigators using Likert-scales in a post-then-pre format [14]. Post-then-pre survey designs are single timepoint surveys asking respondents to reflect after an event has taken place and are useful when participants may not accurately assess their pre-event perceptions (e.g., self-efficacy, engagement) and to account for the response shift bias inherent in traditional pre/post survey designs [14,15]. Two questions related to the activity objectives were not in the post-then-pre format.

Data were analyzed using SPSS for Windows, version 25.0 (IBM Corporation, Armonk, NY, USA). Cronbach’s alpha was used to assess the reliability of the survey instrument. Descriptive statistics were calculated for all variables. Wilcoxon signed ranks tests compared the distribution of responses on each of the post-then-pre survey items. All tests were two-tailed, and an a priori alpha level of 0.05 was considered statistically significant.

## 3. Results

A total of 178 students completed the questionnaire (response rate = 96%). The demographics of the students participating in the survey are presented in Table 1. The mean age of participants was approximately 25 years old.

The Cronbach’s alpha of the evaluation tool was 0.897. Mean scale scores for online post-then-pre self-assessment questionnaire items are shown in Table 2.

For all items, there was a statistically significant difference (*p* < 0.001) in students’ self-rating after the completing the CYOA activity. The activity improved student confidence with evaluating therapeutic options, anticipating outcomes/consequences from different clinical choices, and understanding how the patients’ choices can affect clinical recommendations. Students also described increased comfort adjusting insulin therapy based on glucose log readings. When asked how well the CYOA Activity prepared them to make clinical decisions regarding outpatient intensive insulin therapy, and to understand the patient experience with carbohydrate counting, students responded positively (67.4% and 77.9% responded well or very well, respectively).

The responses to the survey questions relating to student perceptions of the activity are presented in Figure 1. Most students felt that the CYOA activity was engaging (83.7% agree or strongly agree), and 70.8% agreed that more activities like this are needed in similar courses. Additionally, the students reported that this activity helped them to better understand the patient experience (86.5% agree or strongly agree).

## 4. Discussion

The quantitative findings suggest that the CYOA patient case format was successful in increasing self-reported knowledge and confidence related to outpatient diabetes management. As the therapeutic management of diabetes can be a daunting subject, this engaging activity decreased the pressure on students to select the “correct” answer and allowed them to explore clinical options in a low-stakes environment. The students were engaged in the learning activity and able to better understand the patient experience related to outpatient diabetes management and its effect on clinical decision making.

The benefits of technology and educational gaming to enhance student learning have been well described [16]. Online case simulations have also shown increased student engagement compared to traditional paper cases [17]. PBL has also proven to be engaging and has shown increased benefit when compared to online virtual patient cases [18,19]. This CYOA activity incorporates all three active learning concepts to promote student engagement, while also modeling expert thinking to improve critical thinking skills [20]. Positive components of the CYOA patient case format when compared to traditional patient case include its interactivity and visual components. Our data showed that students found the activity engaging, and the activity is constructed so that even students who do not like to speak up in small groups can participate. This activity is conducive for both in-person and virtual learning.

The “choose your own adventure” concept appears to be a promising emerging educational approach in pharmacy. Morningstar-Kywi et al. (2021) evaluated eCases using interactive fiction technology to allow learners to individually explore the narrative of patient presentation, treatment selection, and therapeutic outcomes [21]. Furthermore, Rebitch et al. (2019) administered a similar activity, employing a video case portrayal and modeling of an expert pharmacist’s decision-making approach [22]. While the Rebitch activities are described as “technology enhanced case-based modules” [22], and Morningstar-Kywi utilizes “interactive fiction technology” [21], the CYOA Patient Case format depicted in this study may be best described as a “web-based unfolding patient case scenario” [23]. However, all three examples use technology to strategically break down patient cases and provide feedback, giving students an opportunity to challenge and apply their knowledge in a safe learning environment.

When compared to these studies, the CYOA patient case format requires minimal up-front resources with no video production or technology subscription costs. In our study, the CYOA activity was conducted in small groups to promote discussion, but in the absence of small group facilitators, it may also be completed individually. This activity was previously piloted in fall 2019 for the in-person classroom using hyperlinking in Microsoft PowerPoint [24]; however, it was reformatted for QuestionPro for the virtual-learning environment in response to campus closure due to the COVID-19 pandemic. The benefits of switching to the online survey format included the ability for the instructor to monitor student progress live (through survey response completion) as well as the option to require students to input their rationale for each choice. The rationale for each choice were captured by the investigators and may be used in the future to identify knowledge gaps or areas of opportunity that may be shared with the therapeutics course instructor.

Of note, this activity was administered before the examination on the diabetes material. While this allowed students to “test” their knowledge of class material, this may have influenced their perceptions if they had not yet studied for the exam. Student examination scores were not investigated because the objectives of the IPECS III Course were not the same as the therapeutics course in which they were tested. However, PBL has been shown to improve examination scores in pharmacy education [17,25]. More appropriate endpoints for our future research may include performance on written patient cases, OSCEs or diabetes-related Advanced Pharmacy Practice Experiences (APPEs).

## 5. Strengths and Limitations

While this educational activity was developed specifically for diabetes management, the principle concepts of the CYOA Patient Case Format may be translated to other complex disease states where there are many therapeutic options—such as cardiology, infectious disease, geriatrics—any specialty where clinical expertise and the patient experience must come into play. Future qualitative studies are needed to examine the student experience with this pedagogical tool to develop best practices for the extrapolation of this patient case format to other disease states.

Although this study is limited in that it describes the perceptions of one cohort of students, there is additional support from the pilot of this activity [24]. Additionally, each small group was led by teaching assistant facilitators—this may account for some variability in the student experience.

## 6. Conclusions

The development of a novel CYOA patient case format improved student knowledge and confidence in the management of diabetes. The students consistently rated the interactive class activity highly, and this format shows promise as an educational tool that may be adapted to enhance clinical decision-making skills in other disease states.

## Figures and Tables

**Figure 1 pharmacy-09-00087-f001:**
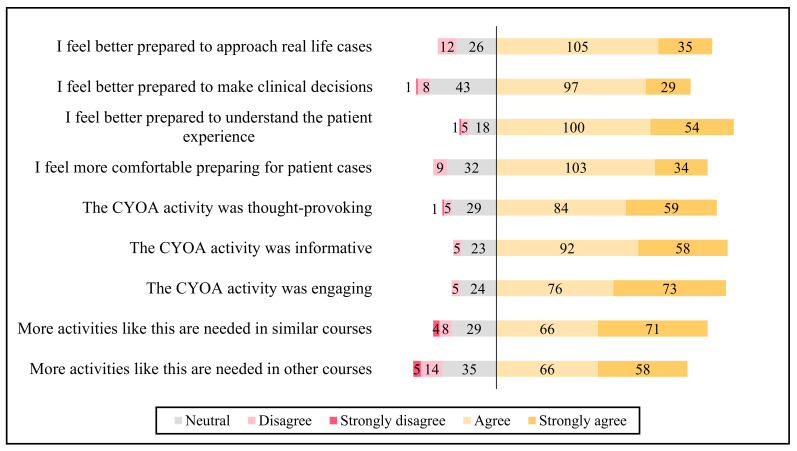
Overall student perceptions of the CYOA activity (*n* = 178).

**Table 1 pharmacy-09-00087-t001:** Demographic characteristics of students who completed the
online post-then-pre self-assessment questionnaire (*n* = 178).

Characteristic		Result
Age, mean (SD)		24.7 (3.62%)
Gender, *n* (%)	Female	115 (64.6%)
	Male	61 (34.3%)
	Prefer not to disclose	2 (1.1%)
Race, *n* (%)	White	111 (62.4%)
	Black/African American	39 (21.9%)
	Asian/Pacific Islander	23 (12.9%)
	Prefer not to disclose	5 (2.8%)
Geographical classification of hometown, *n* (%)	Urban	92 (51.7%)
	Rural	73 (41.0%)
	Other	13 (7.3%)
Prior Bachelor’s degree, *n* (%)		132 (74.2%)
Current pharmacy intern, *n* (%)		123 (69.1%)
Prior pharmacy work experience, *n* (%)		104 (58.4%)

**Table 2 pharmacy-09-00087-t002:** Mean scale scores for online post-then-pre self-assessment questionnaire items (*n* = 178).

	Before the CYOA Activity		After the CYOA Activity		*p* Value ^a^
Questionnaire Item	Mean	SD	Mean	SD	
*(1 = strongly disagree, 2 = disagree, 3 = neutral, 4 = agree, 5 = strongly agree)*					
I feel as though I adequately prepare for patient case studies (reading/reviewing notes)	3.49	0.928	4.06	0.750	<0.001
I feel that I connect pharmacological concepts to patient experiences	3.84	0.743	4.33	0.597	<0.001
I feel that I connect case studies to patient experiences	3.78	0.799	4.28	0.729	<0.001
I feel the case presented in class resonated with me and the real-life situations to which I am/will be exposed when working in clinical pharmacy settings	3.94	0.821	4.34	0.673	<0.001
I feel as though I approach case studies from a patient perspective	3.66	0.870	4.22	0.693	<0.001
*(1 = not confident at all, 2 = not very confident, 3 = neither, 4 = somewhat confident, 5 = very confident)*					
Confidence in critical thinking skills	3.66	0.932	4.19	0.650	<0.001
Confidence in thinking critically about case management.	3.61	0.903	4.16	0.690	<0.001
Confidence in knowing what types of information is most relevant to study for case questions	3.66	0.986	4.23	0.727	<0.001
Confidence in putting multiple pieces of clinical information together	3.67	0.905	4.22	0.717	<0.001
Confidence in evaluating therapeutics options in a clinical case	3.60	0.892	4.18	0.722	<0.001
Confidence in anticipating outcomes/consequences from different clinical choices	3.57	0.913	4.21	0.655	<0.001
Confidence in understanding how the patient’s choices can affect clinical recommendations	3.90	0.851	4.44	0.591	<0.001
*(1 = very uncomfortable, 2 = somewhat uncomfortable, 3 = neither, 4 = somewhat comfortable, 5 = very comfortable)*					
Comfort adjusting insulin therapy based on glucose log readings	3.12	1.111	3.83	0.938	<0.001

^a^ Wilcoxon signed ranks test.

## Data Availability

For any inquiries about the survey, please contact the corresponding author. Due to the privacy of data and in accordance with the IRB approval, the research team cannot share the data.

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
