# Peer review of "Enhancing Student Knowledge of Diabetes through Virtual Choose Your Own Adventure Patient Case Format"

_pharmacy, 2021, doi:10.3390/pharmacy9020087_

Round 1
Reviewer 1 Report
Executive Summary
The manuscript titled “Enhancing Student Knowledge of Diabetes Through Virtual Choose Your Own Adventure Patient Case Format” described research using an interactive Choose Your Own Adventure (CYOA) patient case to train students about pharmacy care for diabetic patients. Further, a brief online questionnaire was filled by students for data collection and analysis. Overall, the manuscript is scientifically written with strong logic. However, the results and discussion do not support the conclusion that “The CYOA activity improved self-reported knowledge of outpatient diabetes management”. Authors may perform minor revisions to better support the conclusion.
Major Comments
- While questionnaires may provide data for “increased self-reported confidence in clinical decision-making skills”, knowledge should not be judged subjectively. Knowledge level should be judged through a well-designed exam or test. Therefore, authors may incorporate an examination on the material, or other data that may serve the same purpose, to better judge the knowledge of students.
Monir Comments
- There is no detail for the interactive Choose Your Own Adventure (CYOA) patient case. It would be helpful to disclose some details, if possible, for reviewers as a reference.
- In Table 1, the authors provide background information including “Current pharmacy intern” and “Prior pharmacy work experience”. Since the goal of this research is to see the teaching results on diabetic care, authors may include information such as “Prior diabetes pharmacy experience”, or other data that may serve the same purpose. If a certain amount of students were exposed to similar training before this class, they may not consider this class as a significant improvement of their confidence or knowledge.
Author Response
Thank you for your time and consideration of this manuscript. Please see attached

Reviewer 2 Report
The paper is overall acceptable. Some minor changes should be made.
Please, form the abstract in the following manner. First, describe the background of the research (1-2 sentences). Second, describe the goals of the research (1-2 sentences). Third, describe briefly (1-2 sentences) the methodology used. Fourth, describe results and the conclusion of the research in 3-4 sentences.
At the end of the introduction, add two paragraphs. In the first paragraph, explain what is the contribution of the paper, in relation to several previous papers - at least 1-2 references from scholarly journals, not later than 2010. In the second paragraph, describe other sections of the paper.
In the last section, please focus on “Conclusion” to include
(1). Conclusions
(2). Academic Implications
(3). Limitations of the paper
(4). Future Studies and Recommendations
Author Response
Thank you for your time and consideration of this manuscript. Please see attached.
